# Comparative Evaluation of a Low-Carbohydrate Diet and a Mediterranean Diet in Overweight/Obese Patients with Type 2 Diabetes Mellitus: A 16-Week Intervention Study

**DOI:** 10.3390/nu16010095

**Published:** 2023-12-27

**Authors:** Walter Currenti, Francesca Losavio, Stefano Quiete, Amer M. Alanazi, Giovanni Messina, Rita Polito, Fabiana Ciolli, Raffaela Simona Zappalà, Fabio Galvano, Raffaele Ivan Cincione

**Affiliations:** 1Department of Biomedical and Biotechnological Sciences, University of Catania, 95123 Catania, Italy; simonazappala@hotmail.com; 2Department of Clinical and Experimental Medicine, University of Foggia, 71100 Foggia, Italy; losaviofrancesca@yahoo.it (F.L.); giovanni.messina@unifg.it (G.M.); rita.polito@unifg.it (R.P.); fabycioll@gmail.com (F.C.); ivan.cincione@unifg.it (R.I.C.); 3Department of Medical and Surgical Sciences, University of Foggia, 71122 Foggia, Italy; stefanoquiete@gmail.com; 4Department of Pharmaceutical Chemistry, College of Pharmacy, King Saud University, P.O. Box 2457, Riyadh 11451, Saudi Arabia; amalanazi@ksu.edu.sa

**Keywords:** low-carbohydrate diet, Mediterranean diet, obesity, type 2 diabetes, body composition, free-fat mass, fat mass, cardiovascular risk, cardiometabolic risk

## Abstract

Introduction: The worldwide prevalence of type 2 diabetes mellitus (T2DM) and obesity has been steadily increasing over the past four decades, with projections indicating a significant rise in the number of affected individuals by 2045. Therapeutic interventions in T2DM aim to control blood glucose levels and reduce the risk of complications. Dietary and lifestyle modifications play a crucial role in the management of T2DM and obesity. While conventional medical nutritional therapy (MNT) often promotes a high-carbohydrate, low-fat Mediterranean diet as an elective treatment, low-carbohydrate diets (LCDs), specifically those restricting carbohydrate intake to less than 130 g/day, have gained popularity due to their multifaceted benefits. Scientific research supports the efficacy of LCDs in improving glycemic control, weight loss, blood pressure, lipid profiles, and overall quality of life. However, sustaining these benefits over the long term remains challenging. This trial aimed to compare the effects of a Mediterranean diet vs. a low-carbohydrate diet (carbohydrate intake < 130 g/day) on overweight/obese patients with T2DM over a 16-week period. The study will evaluate the differential effects of these diets on glycemic regulation, weight reduction, lipid profile, and cardiovascular risk factors. Methods: The study population comprises 100 overweight/obese patients with poorly controlled T2DM. Anthropometric measurements, bioimpedance analysis, and blood chemistry assessments will be conducted at baseline and after the 16-week intervention period. Both dietary interventions were hypocaloric, with a focus on maintaining a 500 kcal/day energy deficit. Results: After 16 weeks, both diets had positive effects on various parameters, including weight loss, blood pressure, glucose control, lipid profile, and renal function. However, the low-carbohydrate diet appears to result in a greater reduction in BMI, blood pressure, waist circumference, glucose levels, lipid profiles, cardiovascular risk, renal markers, and overall metabolic parameters compared to the Mediterranean diet at the 16-week follow up. Conclusions: These findings suggest that a low-carbohydrate diet may be more effective than a Mediterranean diet in promoting weight loss and improving various metabolic and cardiovascular risk factors in overweight/obese patients with T2DM. However, it is important to note that further research is needed to understand the clinical implications and long-term sustainability of these findings.

## 1. Introduction

Over the past forty years, there has been a persistent rise in the worldwide prevalence of type 2 diabetes mellitus (T2DM) and obesity. Projections indicate that by 2045, the number of people with diabetes around the world will increase significantly, with estimates suggesting that approximately 600 million individuals will be affected [1]. Type 2 diabetes mellitus (T2DM) has a significant impact on public health, affecting over 3 million individuals in Italy and is associated with high rates of mortality, disability, and hospitalization, as reported by the ISTAT report on Diabetes in Italy). Therapeutic interventions in T2DM aim to control blood glucose levels and reduce the incidence of both microvascular and macrovascular complications. It has been reported that a 1% reduction in HbA1c levels leads to a 35% decrease in the risk of microvascular complications and reduces the risk of myocardial infarction and sudden death by 16% [2]. Achieving optimal blood glucose levels is a critical therapeutic goal while concurrently addressing concomitant comorbidities, such as dyslipidemia, hypertension, and albuminuria, which are intricately interrelated with the pathophysiology of hyperglycemia [3,4,5]. The modification of dietary and lifestyle factors represents a fundamental tenet in the treatment and management of type 2 diabetes mellitus and obesity [6]. Dietary interventions that institute a substantial negative calorie balance play a critical role in restoring normal pancreatic function and hepatic insulin sensitivity [7,8,9]. Insulin resistance, a significant risk factor for cardiovascular disease in patients with type 2 diabetes (T2DM), is independently linked to ischemic heart disease in T2DM patients. Moreover, T2DM patients who suffer from ischemic heart disease display a greater degree of insulin resistance compared to those without coronary disease, even after controlling for other variables that may influence this association [10,11,12,13]. The conventional medical nutritional therapy (MNT) for managing diabetes typically promotes the adoption of a Mediterranean diet (MD) with high-carbohydrate (50–60% of daily energy requirements) and low-fat regimen (no more than 30% of total energy), emphasizing caloric restriction, increased fiber consumption, and focusing on vegetables, fruits, whole cereals, and legumes, in addition to lean proteins from fish and poultry and healthy fats such as extra virgin olive oil [14,15]. There is growing evidence leading to recommendations from numerous scientific medical societies of diabetology on the use of these dietary approaches to intervene against type 2 diabetes [16,17,18] and overall protection against cardiovascular diseases [19]. However, contemporary literature reviews assessing the comparative effects of carbohydrate restricted diets versus low-fat control diets in overweight/obese patients with type 2 diabetes mellitus have shown divergent and inconclusive results regarding the impact on glycated hemoglobin, weight loss, and cardiovascular risk factors [20,21,22,23,24,25,26]. Dietary fats, particularly saturated fatty acids (SFAs), have been implicated as the cause of the significant rise in obesity and its related diseases [24]. On the contrary, several systematic reviews challenge the link between total SFA intake with cardiovascular disease and recent studies showed that short-chain saturated fatty acids (SCSFAs) especially from dairy foods [27,28,29,30] may exert potential beneficial effects both on metabolic and mental health outcomes [31,32,33].

Thus, the aim of this study was to compare the effects of a Mediterranean diet versus a low-carbohydrate diet (carbohydrate intake < 130 g/day) on overweight/obese patients with T2DM over a 16-week period. The study will evaluate the differential effects of these diets on glycemic regulation, weight reduction, lipid profile, and cardiovascular risk factors.

## 2. Materials and Methods

### 2.1. Study Population

A total number of 100 overweight/obese patients with type 2 diabetes mellitus, with a median duration of diabetes of 5 years (54 females and 46 males, with an average age of 63; BMI 34,4 kg/m^2^; HA1c 8.5%), equally divided in two groups, where one group followed a low-carbohydrate dietary approach (50 patients, 24 men, 26 women) and the other group had a Mediterranean diet (50 patients, 22 men, 28 women) were recruited at the University Service of Dietetic Therapy, Diabetology and Metabolic Diseases, Policlinico Riuniti Hospital of Foggia, Puglia, Italy, as reported in Figure 1. All the women recruited were in the post-menopause stage. The inclusion criteria were age > 18 and < 75 years, BMI > 25 Kg/m^2^, a poorly controlled type 2 diabetes mellitus with HA1c >8.5%, a stable body weight, and no physical activity during the 90 days preceding the study (less than 1.6 METS in 24 h). Furthermore, no physical activity was allowed throughout the study. Exclusion criteria included: pregnancy or lactation, previous gout or hyperuricemia, neoplastic disease, corticosteroids, hypoglycemic therapy with drugs other than metformin, impaired renal function with a serum creatinine ≥1.5 mg/dL, abnormal liver function with alanine aminotransferase and aspartate aminotransferase levels surpassing three times the standard upper limit, heart diseases such as unstable angina, and unstable heart failure. None of the enrolled patients experienced chronic complications of diabetes or previous cardiovascular events. The comorbidities are represented by hypertension and dyslipidemia. Finally, to assess the comparative effects of only two types of diet, i.e., the Mediterranean diet versus a low-carbohydrate diet, all anti-hypertensive, cholesterol lowering, and hypoglycemic drugs were suspended during the study. Before taking part in the study, all subjects provided written permission and it was conducted in accordance with the Declaration of Helsinki.

### 2.2. Study Design

A single-center, prospective, longitudinal, two parallel arm, non-randomized dietary pilot trial was conducted to test the effects of a Mediterranean diet versus a low-carbohydrate dietary approach in overweight/obese patients with uncontrolled T2DM. The study lasted a total of 16 weeks. Anthropometric measurements, bio-impedance analysis, and blood chemistry assessments were performed at the beginning of the study, time 0 (T0), and at the end of the 16 weeks, time 1 (T1).

### 2.3. Dietary Intervention

A registered dietitian provided instructions to participants in both groups regarding the implementation of the two diets. Those assigned to the low-carbohydrate diet (LCD) received a personalized dietary plan with a maximum of 20 E (energy value of the diet)% from carbohydrates (primarily complex and water soluble), 50–60 E% from fat, and 25–30 E% from protein. The consumption of monounsaturated fatty acids (MUFAs) and polyunsaturated fatty acids (PUFAs) was encouraged. During the first visit, they were carefully trained on recognizing foods containing carbohydrates and nutritional education materials were provided. On the other hand, participants allocated to the mediterranean diet (MD) were instructed to adhere to the official Italian society of Diabetology dietary recommendations, which entail 50–60 E% carbohydrates, mainly derived from fruits, vegetables, and whole-grain sources, 20–30 E% fat, where less than 10 E% should be from SFA, and 20–25 E% from protein. For the low-carbohydrate diet, as per definition [29], a quantity of carbohydrates less than 130 g was administered, while for the Mediterranean diet, a quantity of carbohydrates, as per definition, exceeding 45% (about 55%) [15] was administered. Firstly, the individual calorie requirement was calculated by estimating the basal metabolic rate (BMR) through Mifflin-St George equations, which consider the individual’s sex, age, weight, and height of overweight/obese subjects [34]. Total daily energy expenditure (TDEE) was calculated by multiplying the obtained BMR by the lower physical activity factor derived from LARN (Italian Reference Intake Levels of Nutrients and Energy). Both dietary interventions were subjected to a 500 kcal/day energy deficit. Patients have been instructed to have five meals in a day. The typical balanced daily energy distribution was: 20–25 E% at breakfast, 5–10 E% for snacks, 30–35 E% for lunch, and 25–30 E% for dinner. Adherence to the diet was monitored every four weeks by phone calls conducted by trained research staff dietitians. During the telephone calls, patients were asked to report a typical day’s food intake, including details about meals, snacks, and beverages. The researcher engaged in a supportive and motivational discussion with the participants. Patients received guidance on how to overcome specific dietary challenges or navigate social situations that could impact adherence.

### 2.4. Body Composition Assessment

After verifying compliance with the inclusion criteria, anthropometric analysis, bio-electric impedance analysis (BIA), and urine and blood samples were collected before and at the end of the dietary program. The following anthropometric parameters were measured: height, weight, body mass index (BMI = body weight (kg)/height^2^(m)^2^, waist circumference, hips and waist–hip ratio (waist–hip ratio = waist circumference/hips). Through bio-electric impedance analysis (InBody 770. Co., Ltd., Seoul, South Korea), we evaluated the body composition of patients by detecting the following parameters: FM (fat mass) expressed in kilograms, FFM (fat-free mass), muscle mass expressed in kilograms, total body water expressed in kilograms and in percentage.

### 2.5. Urine and Blood Samples

Fasting samples of venous blood and urine were collected and analyzed at the centralized laboratory of the Policlinico Universitario Riuniti di Foggia to evaluate glucose, glycosylated hemoglobin (Hb1Ac), total cholesterol, HDL cholesterol, LDL cholesterol, triglycerides, creatinine, albuminuria, and e-Gfr, calculated using CKD-EPI equations for the glomerular filtration rate [35]. Insulin was measured by the immunochemiluminescent method, blood glucose by the enzymatic method with hexokinase, HbA1c by immunoassay measurement, and total cholesterol, HDL, LDL, and triglycerides by the enzymatic colorimetric assay. Serum creatinine was measured by the enzymatic assay. To determine cardiovascular risk, the cardiovascular risk index (calculated by dividing the total cholesterol values by the HDL cholesterol values) and the cardiovascular (expressed as a percentage of the probability of undergoing a first major cardiovascular event in the next 10 years), were assessed.

### 2.6. Statistical Analysis

We based the power of calculation on the findings of a prior investigation [20], in which the anticipated absolute reduction in HbA1c levels between the LCD and Mediterranean diet cohorts was 0.5%, accompanied by a standard deviation (SD) of 0.408%. Utilizing a two-tailed 5% significance level, a power (1-ß) of 80%, and assuming a follow-up loss of 20%, the calculated sample size required for the study was 80 patients. For the variables considered at T0 and T1 in the two diet groups, the main descriptive statistics (means and standard deviations) were then compiled. The chi-squared test was performed to demonstrate homogeneity between the two groups, while for age the *t*-test for independent samples was performed. To test the difference between T0 and T1 for each variable, paired-sample *t*-tests were performed separately for the two groups. The delta % variable (i.e., delta percent) was constructed to estimate the time differences of each parameter between the start and the end of treatment and assessed according to the following formula: Δ% = [(Z − W)/W] × 100 where Δ% is the percentage variation of each parameter, calculated as the ratio of an absolute margin of variation from the base value. Subsequently, *t*-tests with independent samples were performed to verify the differences between the two diets on delta % variable. The level of significance considered is *p* < 0.01. The analyses were conducted with SPSS v28.

## 3. Results

### 3.1. Baseline Parameters

Except for height and waist–hip ratio, there were no significant differences between the participants in the two dietary groups by age, gender, BMI, blood pressure, lipid profile, renal markers, and glucose homeostasis parameters (fasting glycemia and HbA1c) at baseline as reported in Table 1.

### 3.2. Anthropometric and Body Composition Parameters

Both the low-carbohydrate and Mediterranean diets showed positive effects on weight loss. However, when comparing the effects of the two dietary treatments, the low-carbohydrate diet resulted in a greater reduction in BMI, waist circumference, waist–hip ratio, fat mass, and total body water with no statistically significant differences in free-fat mass, although there is a greater loss of muscle mass in the Mediterranean diet than the low-carbohydrate diet as reported in Table 2. There are no statistically significant gender-related differences in any of the variables considered.

### 3.3. Metabolic Parameters and Cardiovascular Risk Factors 

Positive outcomes were observed in both the low-carbohydrate and Mediterranean dietary approaches on blood pressure, glucose control, lipid profile, cardiovascular risk, and renal function after 16 weeks. In assessing the effects of the dual dietary strategies, it is revealed that the implementation of a low-carbohydrate diet leads to a greater reductions in various health markers including systolic blood pressure, diastolic blood pressure, blood glucose levels, hemoglobin A1c percentage, total blood cholesterol levels, HDL cholesterol levels, cardiovascular index, percentage of cardiovascular risk, LDL cholesterol levels, albuminuria, serum creatinine levels, and e-Gfr over a 16-week period as reported in Table 3. There are no statistically significant gender-related differences in any of the variables considered. To calculate the probability of a first major cardiovascular event in the next 10 years, we used the UKPDS risk engine, which is a specific risk calculator for type 2 diabetes based on 53,000 patients years of data from the UK Prospective Diabetes Study [36,37,38].

### 3.4. Compliance with Treatment

The selection of patients with similar anthropometric (BMI, weight, body composition) and clinical characteristics (years of disease, glucometabolic control status, renal function, dyslipidemia, kidney function, hypertension, and cardiovascular risk), has been predetermined in order to assess the comparative efficacy of the two dietary treatments alone, and thus avoiding any potentially confounding elements. This type of selection and allocation in the two groups has allowed, from the beginning, for the identification and recruitment of the most motivated patients to follow the diet. Preselection before starting the study resulted in achieving total compliance without any dropouts and a homogeneous balance between the two groups in anthropometric and clinical terms.

## 4. Discussion

The current study aimed to assess and compare the effects of a low-carbohydrate diet (LCD) and a Mediterranean diet (MD) on various anthropometric, clinical, body composition, and metabolic parameters in overweight/obese patients with type 2 diabetes mellitus (T2DM). The increasing worldwide prevalence of T2DM and obesity has raised significant public health concerns, with projections indicating a continuing rise in the number of affected individuals and associated morbidity and mortality rates. This underscores the urgent need for effective therapeutic interventions to manage these conditions successfully [39]. Dietary and lifestyle modifications are essential in the treatment and management of T2DM and obesity, making it vital to identify the most effective nutritional strategies to achieve and maintain positive clinical outcomes. Conventional medical nutritional therapy often advocates for the adoption of a Mediterranean diet, which is rich in fruits, vegetables, whole grains, legumes, nuts, seeds, olive oil, and fish, while reducing red meat and sweets. Numerous studies have demonstrated the beneficial effects of adherence to the MD on glycemic parameters in individuals with T2DM. For instance, a cross-over study found that greater adherence to the Mediterranean diet was associated with better glycemic control, including lower HbA1c levels and reduced fasting glucose levels [40]. Additionally, systematic reviews and meta-analyses reported that the MD was effective in reducing fasting blood glucose levels and glycated hemoglobin (HbA1c) in individuals with T2DM [41,42]. The MD’s beneficial effect on diabetes is derived from its nutrient profile, which is particularly rich in fiber, complex carbohydrates, monounsaturated fats, antioxidants, and anti-inflammatory compounds [43,44]. However, the MD diet is a dietary approach characterized by high-carbohydrate intake (50–60% of daily energy requirements at the expense of proteins and fats) and, as known, carbohydrates are the primary macronutrient that significantly affects glycemic control in subjects with diabetes. In this context, a low-carbohydrate diet (LCD) has gained attention as a potential alternative to the Mediterranean diet [45], particularly in the short term, due to its ability to facilitate greater initial weight loss [46,47,48], promote satiety and reduce hunger [49], decrease liver fat content [50,51,52], improve glycemic control and reduce the need for glucose lowering drugs in patients with T2DM [23,53,54,55,56,57].

Although there are numerous studies on the potential beneficial effects of low-carbohydrate diets on overweight diabetic patients, there is a paucity of scientific literature that directly compares the effects of an MD versus an LCD as seen in our study [58]. First, in line with a recent review and meta-analysis of Ajala et al. [59], our study showed that both MD and LCD diets resulted in a significant reduction in weight loss and improvement in blood pressure, glucose control, lipid profile, cardiovascular risk, and renal function after 16 weeks of treatment. This result may be attributable firstly to the caloric restriction induced by both diets (about −500 kcals from TDEE) and the adherence to a structured food plan monitored by qualified personnel. Remarkably, when comparing the effects of the two diets, the low-carbohydrate diet resulted in a greater reduction in several critical parameters at 16 weeks, including BMI, systolic blood pressure, diastolic blood pressure, waist circumference, waist−hip ratio, fat mass, total body water, blood glucose, hemoglobin A1c %, blood cholesterol, HDL cholesterol, cardiovascular index, % cardiovascular risk, LDL cholesterol, albuminuria, serum creatinine levels, and e-Gfr. 

These results may be explained firstly by the fact that low-carbohydrate diets perform better on weight loss in the short term (less than 6 months) especially on visceral adiposity [60]. The reduction in adipose tissue decreases the release of proinflammatory adipokines, such as tumor necrosis factor-alpha (TNF-α) and interleukin-6 (IL-6), which contribute to a low-grade systemic inflammation and insulin resistance [61]. By also reducing carbohydrate intake, LCDs result in decreased glucose availability, leading to lower postprandial glucose excursions and overall blood glucose levels [62]. LCDs thus promote a reduced demand for insulin secretion, facilitating improved glycemic control [63] and reducing glucose toxicity, oxidative stress, and inflammation which can affect beta-cell function [56]. The decreased insulin levels promote the activation of hormone-sensitive lipase (HSL) that facilitates the breakdown of stored triglycerides in adipose tissue, releasing fatty acids into the bloodstream for energy utilization [64,65]. This process contributes to a decrease in circulating triglycerides and an increase in fatty acid oxidation. 

Additionally, low-carbohydrate diets promote the expression of genes involved in lipid metabolism, such as peroxisome proliferator-activated receptor alpha (PPARα), which enhances fatty acid oxidation and contributes to the improvement in blood lipid profiles [66,67]. Moreover, low-carbohydrate diets may enhance the activity of enzymes involved in HDL metabolism, such as lecithin-cholesterol acyltransferase (LCAT), leading to increased HDL cholesterol synthesis and clearance [68]. Therefore, the effects of LCD on blood lipids are mixed [69,70]; it usually appears that LDL cholesterol tends to increase, while triglycerides are drastically lowered and HDL cholesterol rises [71]. A recent randomized controlled trial involving 71 patients with type 2 diabetes showed that LCD does not adversely affect endothelial function and markers of inflammations as interleukin-6 (IL-6) or high-sensitivity C-reactive protein (hsCRP), demonstrating that this dietary approach does not increase the risk of cardiovascular disease [72]. In our study, the LCD group showed a reduction in LDL cholesterol and an increase in HDL, while the decrease in triglycerides levels did not reach statistical significance among groups. Different study results on blood lipids are probably derived from the variable macronutrient composition of low-carbohydrate diets, especially the amount of saturated fats and the carbohydrate reduction. A carbohydrate restricted diet increases the LDL peak particle size, thus becoming less atherogenic and decreases the numbers of total and small LDL particles [73], which may be a more relevant indicator of cardiovascular risk than total LDL cholesterol levels alone [74]. 

Significantly lower insulin levels have also beneficial effects on blood pressure through the reduction in sodium reabsorption [75], norepinephrine [76], and angiotensin II release [77]. Regarding the kidney, insulin has been shown to affect the integrity and function of the glomerular filtration barrier through the synthesis and release of endothelial nitric oxide, a vasodilator, which helps maintain the dilation of the afferent arteriole. This dilation enhances renal blood flow and contributes to the maintenance of the normal glomerular filtration rate (GFR). Insulin also regulates the synthesis and distribution of key components of the glomerular basement membrane, such as type IV collagen and laminin, which are essential for maintaining the integrity of the filtration barrier. Alterations in insulin signaling can disrupt these processes and contribute to glomerular dysfunction and increased permeability, leading to the leakage of albumin into the urine and alterations in creatinine levels [78,79]. 

Our study highlights the potential benefits of a low-carbohydrate diet as a dietary strategy for managing obesity and T2DM. Therefore, to date, our study is one of the first to compare the Mediterranean diet and the moderate low-carbohydrate diet. Considering both anthropometric and clinical parameters, several potential limitations need to be highlighted. First, a limitation of the present study is the small sample size. A larger sample size would have increased the statistical power to detect changes in the variables measured. Second, the duration of the interventions, which is 16 weeks, is a very short period of time to assess the long-term beneficial metabolic effects and sustainability of LCD. Third, there is no consensus on the ideal low-carbohydrate diet for the treatment of T2DM, particularly with regards to the optimal carbohydrate intake for an individual, as it may vary based on factors such as gender, age, and level of physical activity. In scientific literature, a disparity exists in the definition of low-carbohydrate diets as the level of carbohydrate intake varies and the classification of carbohydrates load ranges from moderate to low, making comparisons with other studies difficult. This demonstrates the need for a standard definition and a more rigorous examination of the effects of low-carbohydrate diets on various health outcomes. Nevertheless, to date, there is no shared and common approach. In fact, different dietary strategies for the management of type 2 diabetes are proposed by various international medical scientific associations. The American Diabetes Association (ADA) in 2023 Standards of Medical Care in Diabetes [80], and previously the British Diabetic Association in 2021 position statement, along with the Scientific Advisory Committee on Nutrition (SACN) report on lower carbohydrate diets for adults with type 2 diabetes [81], advocate for the effectiveness of a low-carbohydrate eating pattern in managing patients with type 2 diabetes or prediabetes who are not achieving glucose targets or require glucose lowering medications. Additionally, achieving remission of type 2 diabetes is deemed possible through weight loss and intensive dietary changes [82]. This is supported by a joint consensus statement in 2021 from the American Diabetes Association (ADA), the Endocrine Society, the European Association for the Study of Diabetes, and Diabetes UK [62]. The statement suggests that remission can occur at least six months after initiating a lifestyle intervention and can persist for three months or more without the use of glucose lowering medications, resulting in the maintenance of normal blood glucose levels with HbA1c levels below 6.5% (48 mmol/mol). Conversely, as of the current date, the Italian Society of Diabetology (SID) and the Association of Medical Diabetologists (AMD) recommend a balanced nutritional therapy, such as the Mediterranean diet, instead of a low-carbohydrate approach for the nutritional treatment of type 2 diabetes, as outlined in their 2023 standards of care. The lack of consensus regarding the appropriate low-carbohydrate diet for managing type 2 diabetes mellitus (DM) is apparent. This uncertainty mostly stems from the varying opinions on the most suitable carbohydrate intake for individuals, which can be influenced by characteristics such as gender, age, and degree of physical activity [63]. However, in our 16-week study, a low-carbohydrate diet seems to be superior to the Mediterranean diet for each investigated outcome, but it is necessary to consider the sustainability and long-term compliance of this type of diet. In fact, the restrictive nature of low-carbohydrate diets, particularly those that severely limit carbohydrate intake, may lead to cravings, social challenges, and difficulty in maintaining dietary variety. While low-carbohydrate diets may restrict certain carbohydrate-rich foods, it is crucial to focus on nutrient-rich sources of carbohydrates, such as vegetables, fruits, and whole grains. Adequate intake of essential nutrients, including fiber, vitamins, and minerals, should be prioritized to maintain optimal health while following a low-carbohydrate approach. Finally, the long-term sustainability of these effects and the optimal carbohydrate intake for individual patients warrant further investigation. The findings of this study contribute to the ongoing discussion regarding dietary interventions in the management of T2DM and provide valuable insights for healthcare providers and individuals seeking effective dietary approaches for T2DM management.

## 5. Conclusions

In this study, the results indicate that a low-carbohydrate diet might have superior efficacy compared to a Mediterranean diet in facilitating weight loss and enhancing several metabolic and cardiovascular outcomes among individuals with type 2 diabetes who are overweight or obese. Nevertheless, it is crucial to emphasize that additional research is essential to comprehend the clinical implications and assess the long-term viability of these findings.

## Figures and Tables

**Figure 1 nutrients-16-00095-f001:**
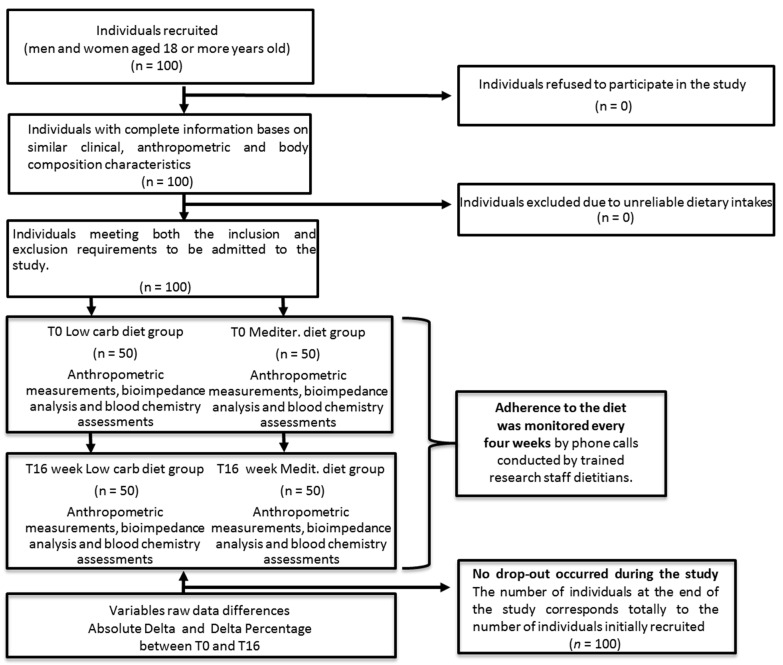
Flowchart of participant selection.

**Table 1 nutrients-16-00095-t001:** Background characteristics of the study population.

	All Patients	Low-Carbohydrate	Mediterranean	
	No. (100)	No. (50)	No. (50)	*p* Value
**Sex, *n* (%)**				
Men	53%	24	22	
Women	47%	26	28	
Age, (SD)	63.2 (6.8)	63.4 (6.9)	63 (6.8)	0.770
Weight, kg (SD)	94.6 (19.7)	91.3 (19.8)	97.9 (19.3)	0.094
Height, m (SD)	1.65 (0.1)	1.63 (0.1)	1.68 (0.1)	0.012
BMI, kg/m^2^ (SD)	34.4 (5.8)	34 (6.2)	34.7 (5.4)	0.575
Systolic blood pressure, mmHg (SD)	126.7 (9.5)	125.7 (9.9)	127.7 (9)	0.294
Diastolic blood pressure, mmHg (SD)	85.5 (8.6)	84.8 (8.9)	86.3 (8.3)	0.387
Waist circumference, cm (SD)	111 (16.3)	113.6 (15.3)	108.4 (16.9)	0.110
Hip circumference, cm (SD)	120.7 (13.4)	118.5 (13.8)	122.9 (12.8)	0.105
Waist–hip ratio, cm (SD)	0.92 (0.1)	0.96 (0.01)	0.88 (0.1)	<0.001 *
Fat mass (FM), kg (SD)	39 (13.8)	36.4 (14.1)	41.7 (13.2)	0.058
Fat-free mass (FFM), kg (SD)	55.9 (9)	55.2 (9.3)	56.6 (8.9)	0.429
Total body water, kg (SD)	40.7 (8.4)	39.8 (8)	41.6 (8.7)	0.297
Blood glucose, mg/dL (SD)	174.7 (47.8)	178 (57.7)	171.3 (35.4)	0.482
Hemoglobin A1c (HbA1c), % (SD)	8.6 (0.7)	8.6 (0.7)	8.6 (0.6)	0.954
Blood cholesterol, mg/dL (SD)	208.6 (21.8)	205.4 (18.9)	211.8 (24.1)	0.144
HDL cholesterol, mg/dL (SD)	45.4 (8.4)	44.5 (8.8)	46.2 (8.1)	0.339
Cardiovascular Index, (SD)	4.75 (0.98)	4.78 (1)	4.72 (0.96)	0.783
% CV risk, (SD)	15.28 (11.9)	14.53 (9.2)	16 (14.1)	0.531
Triglycerides, mg/dL (SD)	173.3 (58)	171.9 (58.3)	174.6 (58.4)	0.815
LDL cholesterol, mg/dL (SD)	128.6 (20.5)	126.5 (17.6)	130.7 (23)	0.306
Albuminuria, mg/dL (SD)	52.1(49.4)	51.1 (50)	53.1 (49.3)	0.840
Serum creatinine level, mg/dL (SD)	0.98 (0.2)	0.99 (0.3)	0.98 (0.2)	0.770
e-Gfr, mL/min (SD)	74 (18.2)	72.4 (18.6)	75.5 (17.8)	0.396

Data are presented as the mean ± standard deviation with * *p* < 0.01. Abbreviations: BMI (body mass index); FM (fat mass); FFM (fat-free mass); HDL (high-density lipoprotein); LDL (low-density lipoprotein); CV (cardiovascular); e-Gfr (estimated glomerular filtration rate).

**Table 2 nutrients-16-00095-t002:** Anthropometric and body composition parameters of the patients at baseline and at the 16-week follow up.

Parameters Mean ± SD	LC Diet	LC Diet *p* Value Baseline to 16 Week	MD Diet	MD *p* Value Baseline to 16 Week	Δ% MD Diet Baseline to 16 Week	Δ% LC Diet Baseline to 16 Week	*p* Value Δ% between Diet Groups
**Weight, kg (SD)** Baseline 16 wk	91.3 (±19.8)	*p* < 0.001	97.9 (±19.3)	*p* < 0.001	−8.20 (±5.7)	−10.1 (±4.3)	0.062
	82.2 (±18.6)		90.3 (±20.8)				
**BMI, kg/m^2^ (SD)** Baseline 16 wk	34 (±6.2)	*p* < 0.001	34.7 (±5.4)	*p* < 0.001	−8.20 (±5.7)	−10.11 (±4.3)	0.062
	30.6 (±5.8)		32 (±6.1)				
**Waist circumference, cm (SD)** Baseline 16 wk	113.6 (±15.3)	*p* < 0.001	108.4 (±16.9)	*p* < 0.001	−6.2 (±4.7)	−8 (±3.7)	0.010
	103.8 (±13.5)		101.9 (±19)				
**Hip circumference, cm (SD)** Baseline 16 wk	118.5 (±13.8)	*p* < 0.001	122.9 (±12.8)	*p* < 0.001	−5.3 (±4.3)	−4.1 (±2.9)	0.101
	113.6 (±12.3)		116.4 (±13.6)				
**Waist–hip ratio, cm (SD)** Baseline 16 wk	0.96 (±0.01)	*p* < 0.001	0.88 (±0.1)	*p* < 0.001	−1 (±3)	−4.6 (±3)	<0.001 *
	0.92 (±0.01)		0.87 (±0.1)				
**Fat mass (FM), kg (SD)** Baseline 16 wk	36.4 (±14.1)	*p* < 0.001	41.7 (±13.2)	*p* < 0.001	−16.5 (±12.4)	−25.4 (±12.8)	<0.001 *
	27.8 (±12.8)		35.6 (±14.6)				
**Fat-free mass (FFM), kg (SD)** Baseline 16 wk	55.2 (±9.3)	*p* < 0.001	56.6 (8.9)	*p* < 0.001	−2.47 (±2)	−1.81 (±3.7)	0.269
	54.2 (±9.3)		55.2 (8.9)				
**Total body water, kg (SD)** Baseline 16 wk	39.8 (±8)	*p* < 0.001	41.6 (±8.7)	*p* < 0.001	−3.4 (±2.6)	−2 (±3.4)	0.022
	39 (±7.9)		40.2 (±8.7)				

Data are presented as the mean ± standard deviation and as Δ% with * *p* < 0.01. Abbreviations: BMI (body mass index); FM (fat mass); FFM (fat-free mass); LC (low-carbohydrate diet); MD (Mediterranean diet).

**Table 3 nutrients-16-00095-t003:** Metabolic parameters and cardiovascular risk factors of the patients at baseline and at the 16-week follow up.

Parameters Mean ± SD	LC Diet	LC Diet *p* Value Baseline to 16 Week	MD Diet	MD Diet *p* Value Baseline to 16 Week	Δ% MD Diet Baseline to 16 Week	Δ% LC Diet Baseline to 16 Week	*p* Value Δ% between Diet Groups
**Blood glucose, mg/dL** (SD) Baseline 16 wk	178 (±57.7)	*p* < 0.001	171.3 (35.4)	*p* < 0.001	−21 (±7.9)	−37.6 (±13.9)	<0.001 *
	104.7 (±17.9)		134.7 (±28.4)				
**Hemoglobin A1c (HbA1c)**, % (SD) Baseline 16 wk	8.6 (±0.7)	*p* < 0.001	8.6 (±0.6)	*p* < 0.001	−13.5 (±3.2)	−23.2 (±5)	<0.001 *
	6.6 (±0.7)		7.4 (±0.6)				
**Blood cholesterol, mg/dL (SD)** Baseline 16 wk	205.4 (±18.9)	*p* < 0.001	211.8 (±24.1)	*p* < 0.001	−13 (±8.6)	−18 (±6.6)	0.002
	168.1 (±18)		182.9 (±17.4)				
**HDL cholesterol**, mg/dL (SD) Baseline 16 wk	44.5 (±8.8)	*p* < 0.001	46.2 (±8.1)	*p* < 0.001	7.1 (±7)	12.8 (±13.4)	0.009
	50 (±10.2)		49.2 (±7.9)				
**Triglycerides, mg/dL** (SD) Baseline 16 wk	171.9 (±58.3)	*p* < 0.001	174.6 (±58.4)	*p* < 0.001	−36.9 (±18.9)	−41 (±17.7)	0.268
	96.7 (±30.1)		106.5 (±43.5)				
**LDL cholesterol, mg/dL (SD)** Baseline 16 wk	126.5 (±17.6)	*p* < 0.001	130.7 (±23)	*p* < 0.001	−12.6 (±14.2)	−21.6 (±12.5)	0.001
	98.7 (±19.1)		112.4 (±19)				
**Albuminuria, mg/dL (SD)** Baseline 16 wk	51.1 (±50)	*p* < 0.001	53.1 (±49.3)	*p* < 0.001	−21.7 (±16.1)	−48.4 (±25.6)	<0.001 *
	26.3 (±27.9)		41.6 (±40.6)				
**Serum creatinine level, mg/dL (SD)** Baseline 16 wk	0.99 (±0.25)	*p* < 0.001	0.98 (±0.23)	*p* < 0.001	−5.7 (±4.9)	−22.3 (±8.8)	<0.001 *
	0.76 (±0.15)		0.92 (±0.20)				
**e-Gfr, mL/min (SD)** Baseline 16 wk	72.4 (±18.6)	*p* < 0.001	75.5 (±17.8)	*p* < 0.001	6.5 (±6.1)	29.7 (±22.7)	<0.001 *
	90.4 (±12.4)		79.8 (±16.9)				
**Cardiovascular Index, (SD)** Baseline 16 wk	4.78 (±1)	*p* < 0.001	4.72 (±0.96)	*p* < 0.001	−18.3 (±10.8)	−26.4 (±10)	<0.001 *
	3.5 (±0.8)		3.83 (±0.88)				
**% CV risk, (SD)** Baseline 16 wk	14.53 (±9.2)	*p* < 0.001	16 (±14.1)	*p* < 0.001	−20.2 (±8.9)	−32.6 (±14.7)	<0.001 *
	9.5 (±6.5)		12.9 (±11.7)				
**Systolic blood pressure, mmHg (SD)** Baseline 16 wk	125.7 (±9.9)	*p* < 0.001	127.7 (±9)	*p* < 0.001	−4.2 (±3)	−7 (±4.2)	<0.001 *
	116.6 (±6)		122.1 (±6)				
**Diastolic blood pressure, mmHg (SD)** Baseline 16 wk	84.8 (±8.9)	*p* < 0.001	86.3 (±8.3)	*p* < 0.001	−4.62 (±3.7)	−9.15 (±5)	<0.001 *
	76.7 (±5.5)		82.1 (±6.2)				

Data are presented as the mean ± standard deviation and as Δ% with * *p* < 0.01. Abbreviations: HDL (high-density lipoprotein); LDL (low-density lipoprotein); e-Gfr (estimated glomerular filtration rate); CV (cardiovascular).

## Data Availability

The data that support the findings of this study are available upon reasonable request.

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
