# Peer review of "Comparative Evaluation of a Low-Carbohydrate Diet and a Mediterranean Diet in Overweight/Obese Patients with Type 2 Diabetes Mellitus: A 16-Week Intervention Study"

_nutrients, 2023, doi:10.3390/nu16010095_

Round 1

Reviewer 1 Report

Comments and Suggestions for Authors

I read this article with interest because there is still controversy in diabetology about the macronutrient content and there is no consensus on the amount of carbohydrates in the diet. There is growing body of evidence for the effectiveness of a low-carbohydrate diet in the management of type 2 diabetes and obesity.

The authors presented interesting reports on the effect of low-carbohydrate diets on improving metabolic parameters in patients with type 2 diabetes. 

My comments on the manuscript:

In the introduction: 

Microalbuminuria - this term is no longer used - instead of microalbuminuria we use albuminuria 

There is an inconsistency in writing type 2 diabetes in the article - Type 2 diabetes - with Arabic numerals rather than Roman numerals

Inclusion criteria:

How long have patients suffered from diabetes? What chronic complications of diabetes and comorbidities did they have? Were they taking flozins for cardiovascular indications, which may affect body weight and metabolic parameters.

HbA1c 8.5% only? Rather it should be > 8.5%

What about weight stability before inclusion in the study? Patients should have had a stable body weight for a minimum of 3 months prior to inclusion or a fluctuation of less than 5% of baseline body weight.

What about the physical activity of the patients? Physical activity has an impact on body weight and lean body mass. Was it controlled in anyway?

How was the individual calorie requirement for the patient determined?

Did patients undergo education on macronutrient counting in diet?

How many meals did they consume?

How was the nutritional intervention supervised?

Presentation of the results: 

Was the distribution of variables checked? 

At first glance, body weight is significantly different in the two groups: 91.3 (19.8) vs 97.9 (19.3) - am I right?

What scale was used to calculate the probability of a first major cardiovascular event in the next 10 years in patients with t. 2 diabetes?

What method is used to calculate eGFR?

What does the abbreviation E% mean?

In the Tables, please standardise the figures - please give data to 1 decimal place

In the discussion, please outline the limitations of the study.

Comments on the Quality of English Language

Minor editing of English language required

Author Response

REVIEWER 1

I read this article with interest because there is still controversy in diabetology about the macronutrient content and there is no consensus on the amount of carbohydrates in the diet. There is growing body of evidence for the effectiveness of a low-carbohydrate diet in the management of type 2 diabetes and obesity. The authors presented interesting reports on the effect of low-carbohydrate diets on improving metabolic parameters in patients with type 2 diabetes.  My comments on the manuscript:

In the introduction: 

  1. Microalbuminuria - this term is no longer used - instead of microalbuminuria we use albuminuria

Author response; thanks for the advice we changed microalbuminuria in to albuminuria

  1. There is an inconsistency in writing type 2 diabetes in the article - Type 2 diabetes - with Arabic numerals rather than Roman numerals

 Author response; sorry for the inattention. We substitute roman numbers with arabic numbers thanks

  1. How long have patients suffered from diabetes? What chronic complications of diabetes and comorbidities did they have? Were they taking flozins for cardiovascular indications, which may affect body weight and metabolic parameters.

 Author response: 

The median duration of diabetes was 5 years. None of the enrolled patients experienced chronic complications of diabetes or previous cardiovascular events.The only two comorbidities were represented by hypertension and dyslipidemia. To assess the comparative effects of only two types of diet, i.e. the Mediterranean diet versus a low-carbohydrate diet, all antihypertensive,cholesterol-lowering and hypoglycemic drugs were suspended during the study.

We added in the “ Materials and methods under Study Population”, thanks

  1. HbA1c 8.5% only? Rather it should be > 8.5%

Author response; sorry for the inattention. It was >8.5% 

  1. What about weight stability before inclusion in the study? Patients should have had a stable body weight for a minimum of 3 months prior to inclusion or a fluctuation of less than 5% of baseline body weight.

Author response; Yes, the patients had a stable weight at least in the 3 months prior to the study and none of them were following a diet. We added in the inclusion criteria thanks

  1. What about the physical activity of the patients? Physical activity has an impact on body weight and lean body mass. Was it controlled in anyway?

Author response; Regarding the physical activity of patients, the study required as inclusion criteria that individuals must be sedentary and did not practice any structured physical activity,  for 90 days preceding the study and during the entire duration of the study. All patients declared to be sedentary and that they do not practice any structured physical activity or heavy job activities. We have highlighted this more clearly in the“ Materials and methods under Study Population”, thanks

  1. How was the individual calorie requirement for the patient determined?

Author response; Individual calorie requirement was calculated firstly estimating basal metabolic rate (BMR) through Mifflin-St George equations. Total daily energy expenditure (TDEE) was calculated by multiplying the BMR by the lower physical activity factor derived from LARN (Italian Reference Intake Levels of Nutrients and Energy). Finally 500 kcal were subtracted from the TDEE in order to obtain hypocaloric diets. We added these details on the paper in “dietary intervention” thanks.

  1. Did patients undergo education on macronutrient counting in diet?

Author response; there was probably a misunderstanding due to which we wrote “Those assigned to the low-carbohydrate diet (LCD) were advised to structure their dietary intake”. Patients received personalized dietary plans and, in addition during the first visit, training on recognition foods containing carbohydrates. So we do not give any carbohydrate counting to be done by the patients themselves, diets had already been developed and balanced by us.

  1. How many meals did they consume?

Author response;  patients have been instructed to have five meals in a day. The typical balanced daily energy distribution was: 20-25 E% at breakfast, 5-10 E% for snacks, 30-35 E% for lunch and 25-30 E% for dinner. We added these details on the paper in “dietary intervention” thanks.

  1. How was the nutritional intervention supervised?

Author response: Adherence to the diet was monitored every four weeks by phone calls conducted by trained research staff dietitians. During the telephone calls patients were asked to report a typical day's food intake, including details about meals, snacks, and beverages. The researcher engaged in a supportive and motivational discussion with participants. Patients received guidance on how to overcome specific dietary challenges or navigate social situations that could impact adherence.

Presentation of the results: 

  1. Was the distribution of variables checked? 

Author response:in order to conduct statistical analysis, all individual variables were necessarily and previously checked to determine whether their distribution was normal or not, thanks

  1. At first glance, body weight is significantly different in the two groups: 91.3 (19.8) vs 97.9 (19.3) - am I right?

Author response: as reported in table 1 the body weight difference between the two groups had p-value = 0.094, while the statistical significance had a p-value < 0.01,thanks    

  1. What scale was used to calculate the probability of a first major cardiovascular event in the next 10 years in patients with t. 2 diabetes?

Author response: as the risk calculators based on equations from the Framingham Heart study tend to underestimate the risks for people with diabetes while this study included relatively few diabetic subjects. So in our study we used the UKPDS Risk Engine, that is a specific risk calculator for type 2 diabetes based on 53,000 patients years of data from the UK Prospective Diabetes Study,thanks 

- Kanaya et al. Explaining the sex difference in coronary heart disease mortality among patients with    

  type 2 diabetes mellitus: a meta-analysis. Arch Intern Med. 2002;162:1737-45.

- Stevens et al. The UKPDS risk engine. Clinical Science 2001:101:671-679

-The UK Prospective Diabetes Study (UKPDS): clinical and therapeutic implications for type 2 diabetes. Paromita King, Ian Peacock, Richard Donnelly December 2001 https://doi.org/10.1046/j.1365-2125.1999.00092.x )

  1. What method is used to calculate eGFR?

Author response; e-GFR was calculated using CKD-EPI Equations for Glomerular Filtration Rate,thanks

(Levey AS, Stevens LA, Schmid CH, Zhang YL, Castro AF 3rd, Feldman HI, Kusek JW, Eggers P, Van Lente F, Greene T, Coresh J; CKD-EPI (Chronic Kidney Disease Epidemiology Collaboration). A New Equation to Estimate Glomerular Filtration Rate. Ann Intern Med 150(9):604-12. (2009))

  1. What does the abbreviation E% mean?

Author response; percentage from Energy value of the diet. We add in the text thanks

  1. In the Tables, please standardise the figures - please give data to 1 decimal place

Author response; we standardized as suggested thanks.

  1. In the discussion, please outline the limitations of the study.

Author response; thanks we followed the advice

Reviewer 2 Report

Comments and Suggestions for Authors

1.This study was non-randomized trial among Italian adults to assess the association between dietary fat intake and depressive symptoms. The authors would need to provide more details in this study.  In particular, the authors should provide a flowchart of participant selection, in which how the final study sample was determined.

2.The sample of this study was 100 overweight patients (Line 91), but LCD and MD group included 50 and 40 patients (Line 94). I think the authors wrote wrong by mistake. Please check it carefully.

3.In recruited patients, participants who take the hypoglycemic were included. Would medication affect outcome measures such as blood glucose, lipids and blood pressure?

4.Why is age group used instead of age in Table1? It seems that the mean age (63) is more like a result of continuous variables.

5.Compliance is very important in intervention studies, and it is not mentioned in this article how to ensure the compliance of study participants. The authors need to add results on participants compliance.

6.I would like to know why the authors only listed △% and did not list the change value and test it statistically. The group × time interaction effect should also be examined.

Comments on the Quality of English Language

1.The authors should check the position of the punctuation, such as Line 94.

2.The authors need to confirm whether a decimal point or a comma is used in Table1.

3. The data in the article should be kept in the same number of decimal places.

4.All significant P-value should be marked with "*".

5.The discussion should be segmented according to different aspects, not a whole paragraph.

Author Response

REVIEWER 2

1.This study was a non-randomized trial among Italian adults to assess the association between dietary fat intake and depressive symptoms. The authors would need to provide more details in this study.  In particular, the authors should provide a flowchart of participant selection, in which how the final study sample was determined.

Author response;  thanks we followed the advice, a flowchart of participant selection has been added thanks

2.The sample of this study was 100 overweight patients (Line 91), but the LCD and MD group included 50 and 40 patients (Line 94). I think the authors wrote wrong by mistake. Please check it carefully.

Author response: sorry for the inattention we corrected thanks

3.In recruited patients, participants who take the hypoglycemic were included. Would medication affect outcome measures such as blood glucose, lipids and blood pressure?

Author response: to assess the comparative effects of only two types of diet, i.e. the Mediterranean diet versus a low-carbohydrate diet, all antihypertensive,cholesterol-lowering and hypoglycemic drugs were suspended during the study. We added in the Materials and methods under Study Population, thanks

4.Why is age group used instead of age in Table1? It seems that the mean age (63) is more like a result of continuous variables.

Author response: sorry for the inattention we corrected thanks

  1. Compliance is very important in intervention studies, and it is not mentioned in this article how to ensure the compliance of study participants. The authors need to add results on participants' compliance.

Author response: Adherence to the diet was monitored every four weeks by phone calls conducted by trained research staff dietitians. During the telephone calls patients were asked to report a typical day's food intake, including details about meals, snacks, and beverages. The researcher engaged in a supportive and motivational discussion with participants. Patients received guidance on how to overcome specific dietary challenges or navigate social situations that could impact adherence.No drop-out occurred during the study and the number of individuals at the end of the study corresponds totally to the number of individuals initially recruited ( 100/100),thank

6.I would like to know why the authors only listed △% and did not list the change value and test it statistically. The group × time interaction effect should also be examined.

Author response: The delta % variable (i.e. delta percent)  was used to estimate the time differences of each parameter of each of two groups  between the start and the end of treatment, as the percentage variation of each parameter, calculated as the ratio of an absolute margin of variation from the base value.It is our opinion that the delta % variable allows to capture and to represent in a more appropriate,immediate and clinically useful way the changes occurred, with respect to the use of absolute values. It also expresses the time dimension (T0-T1) related to all variables of each of the two diet groups (Mediterranean diet versus low carb ).Ultimately delta percent is a complete and versatile index, thanks  

1.The authors should check the position of the punctuation, such as Line 94.

Author response: sorry for the inattention we checked punctuation. We also checked English as requested.

2.The authors need to confirm whether a decimal point or a comma is used in Table1.

Author response: sorry for the inattention we use only decimal point

  1. The data in the article should be kept in the same number of decimal places.

Author response; we standardized as suggested thanks.

4.All significant P-value should be marked with "*".

Author response: thanks we marked as requested

5.The discussion should be segmented according to different aspects, not a whole paragraph.

Author response: thanks for the advice we separated the paragraphs

Reviewer 3 Report

Comments and Suggestions for Authors

This article discusses the effect of two types of diet (low-carbohydrate and Mediterranean) on weight loss and improvement of cardiometabolic risk in obese patients with type 2 diabetes.

The subject is quite topical since obesity is currently a pandemic that requires effective interventions to try to control it. It is interesting to know the efficacy of these two diets to try to reduce obesity. Furthermore, there are not too many studies comparing these two types of diets, which is why this study is of special relevance.

The study follows an adequate methodology but the sample size (50 people in each group) seems a little low and also the follow-up time (16 weeks) is too short to evaluate the long-term effect of these two diets.

The conclusions obtained in the study are coherent with the analyses carried out and with the objectives and the main question posed.

The bibliographic references, although adequate, are somewhat obsolete, since the obsolescence rate is 8 years. Only 36.5% of the references are less than 5 years old and 17.46% are less than two years old.

The tables are adequate and facilitate reading and understanding of the text.

Author Response

REVIEWER 3

This article discusses the effect of two types of diet (low-carbohydrate and Mediterranean) on weight loss and improvement of cardiometabolic risk in obese patients with type 2 diabetes.

The subject is quite topical since obesity is currently a pandemic that requires effective interventions to try to control it. It is interesting to know the efficacy of these two diets to try to reduce obesity. Furthermore, there are not too many studies comparing these two types of diets, which is why this study is of special relevance. The study follows an adequate methodology but the sample size (50 people in each group) seems a little low and also the follow-up time (16 weeks) is too short to evaluate the long-term effect of these two diets. The conclusions obtained in the study are coherent with the analyses carried out and with the objectives and the main question posed.

  1. The bibliographic references, although adequate, are somewhat obsolete, since the obsolescence rate is 8 years. Only 36.5% of the references are less than 5 years old and 17.46% are less than two years old. The tables are adequate and facilitate reading and understanding of the text.

Author response:  first of all thanks for your consideration. We agree that 100 patients are few and that the duration of the study is short therefore we have included it in the limitations of the study.  As suggested we have added more recent bibliographical references PMID: 38027413; PMID: 38004162; PMID: 37966583; PMID: 37950240; PMID: 37875472; PMID: 37862824; PMID: 37769437; PMID: 37701967; PMID: 37592243; PMID: 37513574; PMID: 37317478; PMID: 37278294; PMID: 37559961. thanks

Reviewer 4 Report

Comments and Suggestions for Authors

Thank you for inviting me to review your work titled: Comparative Evaluation of a Low-Carbohydrate Diet and a Mediterranean Diet in Overweight/Obese Patients with Type 2 Diabetes Mellitus: A 16-Week Intervention Study. I suggest applying the following changes:

1.       Use the term: type 2 diabetes mellitus,  and not: type II diabetes mellitus

2.       In the Introduction section, clearly write the purpose of the work

3.       Present (e.g., in a table) the characteristics of laboratory parameters of patients' blood and urine samples (before and after the diet), including gender (premenopausal and postmenopausal women), drugs used and their dosages

4.       Describe the methods used to determine biochemical parameters in blood and urine

5.       Describe your results by gender of patients

Author Response

REVIEWER 4

Thank you for inviting me to review your work titled: Comparative Evaluation of a Low-Carbohydrate Diet and a Mediterranean Diet in Overweight/Obese Patients with Type 2 Diabetes Mellitus: A 16-Week Intervention Study. I suggest applying the following changes:

  1.       Use the term: type 2 diabetes mellitus,  and not: type II diabetes mellitus

Author response: sorry for the inattention. We substitute as requested

  1.       In the Introduction section, clearly write the purpose of the work

Author response: Thanks we added at the end of the introduction 

  1.       Present (e.g., in a table) the characteristics of laboratory parameters of patients' blood and urine samples (before and after the diet), including gender (premenopausal and postmenopausal women), drugs used and their dosages

Author response: The characteristics of laboratory parameters of patients' samples at baseline and after 16 weeks are reported in Table 3 (before and after the diet). All the women  recruited were in the postmenopausal stage, as reported in the “ Materials and methods under Study Population”, There are no statistically significant gender-related differences in any of the variables considered in the study .To assess the comparative effects of only two types of diet, i.e. the Mediterranean diet versus a low-carbohydrate diet, all antihypertensive, cholesterol-lowering and hypoglycemic drugs were suspended during the study,thanks

  1.       Describe the methods used to determine biochemical parameters in blood and urine

Author response: insulin, was measured by the immunochemiluminescent method, blood glucose by the enzymatic method with hexokinase, HbA1c by immunoassays measure, total cholesterol, HDL, LDL and triglycerides by the enzymatic colorimetric assay. Serum creatinine was measured by enzymatic assay, thanks

  1.       Describe your results by gender of patients

Author response: There are no statistically significant gender-related differences in any of the variables considered in the present  study. Gender-related differences were statistically verified at the beginning of the data analysis. The absence of gender differences made it possible to use the total sample of individuals as a whole without gender distinction, thanks

Round 2

Reviewer 2 Report

Comments and Suggestions for Authors

1, Results of compliance need to be reported.

2, Since allocation was not randomly conducted. How did the allocation was carried out? who decided?

3, In the study flow, information during and after intervention was missing. 

Author Response

1) Results of compliance need to be reported.

Author: thanks for the advice and sorry for the inattention. As requested we added the 3.4 Compliance with treatment in the results section

2) Since allocation was not randomly conducted. How did the allocation was carried out? who decided?

Author: the selection of patients with similar anthropometric ( bmi, weight, body composition) and clinics characteristics ( years of disease, glycometabolic control status, renal function, dyslipidemia, kidney function, hypertension, and cardiovascular risk), has been predetermined in order to assess the comparative efficacy of the two dietary treatments alone, thus avoiding any potentially confounding elements. Moreover, this type of selection and allocation in the two groups, has allowed to identify and then recruit from the beginning only the most motivated patients to follow the diet. We achieved a total compliance without any drop-out and a homogeneous balance between the two groups in anthropometric and clinical terms, as the result of pre-selection already before the start of the study. We added this text in the 3.4 Compliance with treatment  section under the results section. 

3) In the study flow, information during and after intervention was missing.

Author: thanks for the advice and sorry for the inattention. We updated the flow chart

Reviewer 4 Report

Comments and Suggestions for Authors

Accept in present form

Author Response

thanks